# Field Drying for Enhancing Biomass Quality of Eucalyptus Logs and Trees in Florida, USA

**Donald L. Rockwood** [1,*] **, Bijay Tamang** [2] **and Martin F. Ellis** [3]

1 School of Forest, Fisheries, and Geomatics Sciences, University of Florida, Gainesville, FL 32611, USA
2 F4 Tech, Tallahassee, FL 32301, USA
3 Green Carbon Solutions, Pepper Pike, OH 44124, USA
* Correspondence: dlrock@ufl.edu; Tel.: +1-352-256-3474

**Abstract:** Field drying *Eucalyptus* biomass can minimize the storage requirements, transportation costs, and/or the cost associated with biomass drying for biochar and other products. In central Florida, 12 6-year-old and 12 8-year-old *Eucalyptus amplifolia* and *Eucalyptus grandis* trees were field dried over two seasons, with two months of field drying in each season, and two tree forms (logs, whole tree). The whole tree average initial moisture content ($MC_{od}$) on a dry basis ranged between 115 and 121% in *E. amplifolia* and 116 and 119% in *E. grandis*. The season and duration of field drying affected drying, but species, tree size, and tree form did not. In southern Florida, 19 3+-year-old coppice stems of an *E. grandis* × *Eucalyptus urophylla* hybrid clone (EH1) were field dried over two months of one season, with and without tree tops. The whole tree initial $MC_{od}$ ranged between 119 and 138%. The duration of field drying affected drying, but tree size and tree tops did not. Both studies suggest that field drying can effectively reduce wood MC and thus enhance *E. amplifolia*, *E. grandis*, and EH1 biomass quality.

**Keywords:** *Eucalyptus*; woody biomass; biomass quality; moisture content; transpirational drying

## 1. Introduction

Woody biomass has received worldwide attention for energy production which can significantly reduce dependency on fossil fuel. Woody biomass conversion involves biochemical and thermochemical (directly-fired or conventional steam approach, co-firing, pyrolysis, and gasification) processes. Both processes require moisture content (MC) at different levels. While biomass with >50% MC is ideal for biological conversion, depending on the conversion method, woody biomass MC needs to be 5%–20% for thermochemical processes [1–3]. As woody biomass generally is harvested, processed, and transported for immediate use, the period between harvest and conversion is too short for the biomass to dry naturally. A higher MC can decrease the efficiency of coal-fired power plants when biomass is co-fired [4]. A higher MC also increases the capital cost as larger equipment (e.g., boilers) are needed to convert biomass with higher MC levels [5]. On the other hand, a low biomass MC increases the net energy yield [6]. Therefore, the biomass either needs to be stored until achieving the desired MC or dried before it can be used, which requires time and/or energy. A lower MC is also one of the many ideal wood properties for ethanol and methanol production regardless of the technology used [7]. Transpirational, i.e., field, drying is one of the most economical methods to reduce biomass MC and produce significant heat energy gains [8,9].

The end-product and process requirements influence the desired MC in the thermochemical conversion of biomass. For example, the pyrolysis process may be optimized for biochar, pyrolytic liquids, or syngas and energy. Depending on the optimization objective, higher or lower MCs may be desirable [10]. MC will also influence the economics of the end-product, and research reveals that a low MC after field drying produces savings in

energy during feedstock transportation and on-site drying [11]. Irrespective of the end-product objective, there are significant thermochemical production benefits to using a mono feedstock with a consistent MC.

While several woody species can be used for bioenergy, *Eucalyptus* is uniquely positioned as a feedstock given its fast growth, consistency, and relatively high density. When these woody biomass features are combined with a low moisture content as a result of effective field drying, there can be significant product benefits.

Of the many *Eucalyptus* species, the world's most widely planted hardwoods [12], *E. grandis* Hill ex Maiden, *E. amplifolia* Naudin, and *E. grandis* × *E. urophylla* S.T. Blake hybrids are well suited to subtropical central and southern Florida, USA, because of their fast growth, short rotation, high productivity, and tolerance of various site conditions, and they have numerous potential applications whether grown in long rotations or as short rotation woody crops (SRWC). Through genetic improvement, high planting density, and intensive site amendments such as biochar [13], *E. grandis* cultivars may have maximum mean annual increments up to 78.2 green $Mg\ ha^{-1}\ year^{-1}$ and sequester over 10 Mg of $C\ ha^{-1}\ year^{-1}$ when grown as SRWCs [14]. Biochar increases soil carbon and productivity [15–20], which is especially important for the sandy soils common to central and southern Florida [21]. Soil amendment using biochar derived from eucalypts may therefore both enhance long-term carbon sequestration and increase plantation productivity [22].

Despite its high productivity, there is some environmental concern about *Eucalyptus* because of its high water use and transpiration rate. Comparative studies have shown that water use by *Eucalyptus* and *Pinus* is not significantly different under certain conditions [23,24]. A recent study has shown water use by *Eucalyptus* to be more efficient than *Pinus* [25]. A part of the water taken from the soil through the roots is lost through transpiration and a part is retained in different parts of the tree. Water is stored in wood in two forms: free water and bound water [26]. Free water is present in liquid and vapor form in the lumen and voids in wood above the fiber saturation point (FSP). Bound water, which is below FSP, is held in cell walls in vapor form by intermolecular forces.

Because of the variation in forces involved in the storage of free and bound water, their movement in the wood is different. The movement of free water is caused by capillary forces whereas the movement of bound water through a cell wall is caused by diffusion as a result of the moisture gradient [27]. During the drying process, free water is removed early on at a higher rate usually with less energy until the FSP is reached. After the FSP is reached, more energy is needed to remove the bound water. Wood will continue to lose moisture until an equilibrium is reached with ambient moisture. Moisture at this point is called equilibrium moisture content (EMC), after which further moisture loss from wood will be difficult without energy input.

The MCs of *E. grandis* and *E. amplifolia* in Florida have been recorded as high as 131% [9] and 108% [28], respectively. Because wood MC reduction could significantly enhance woody biomass quality for energy production and at the same time reduce drying energy costs, we assessed field drying as a potentially cost-effective technique to minimize storage and/or processing requirements. The specific main objectives were to estimate the effects of season, the duration of drying, the species, the tree form, and the tree size for reducing wood MC.

## 2. Materials and Methods

The two studies contributing to assessing these effects represented three species and two management scenarios. Trees in the Florida Organics Recycling Center for Excellence (FORCE) study were older, larger seed-origin *E. amplifolia* and *E. grandis* in a longer first-rotation, and the trees in the Indian River Research and Education Center (IRREC) study were younger, smaller cultivars of *E. grandis* × *E. urophylla* (EH1) in a shorter coppice rotation.

## 2.1. FORCE Study

The FORCE drying study beginning in February 2010 utilized existing FORCE field studies 102 and 102A near Sumterville, Florida (28°44′51″ N, 82°5′17″ W). Average maximum and minimum temperatures at FORCE are 27.2 and 15.6 °C, respectively, and average annual rainfall is 1261 mm. Study 102 included three species and four silvicultural treatments in a split-plot, randomized complete block design to evaluate the effect of compost on tree growth (Table 1). Trees were planted 1 m apart in 14 rows that were 3 m apart and in two rows that had paired trees planted 0.8 m apart. Study 102A, established west of Study 102, further evaluated the effect of compost on four species under four treatments with trees planted 1 m apart in 14 paired rows (0.8 m apart) on 4.3 m centers.

**Table 1.** Planting date, species, planting density (trees ha$^{-1}$), and silvicultural treatments for the FORCE and IRREC studies.

| Study | Date | Species | Density | Treatments |
|---|---|---|---|---|
| FORCE-102 | 09/2002 | *E. amplifolia*, *E. grandis*, *Populus deltoides* Bartr. | 3333, 6666 | Control, compost, irrigation, compost + irrigation; |
| FORCE-102A | 04/2004 | *E. amplifolia*, *E.grandis*, *P. deltoides*, *Taxodium distichum* (L.) Rich | 4651 | Control, compost, irrigation, irrigation + fertilizer, irrigation + compost |
| IRREC | 06/2015 | *E. grandis* × *E. urophylla* | 1111, 1667, 3333 | Control, four fertilizers |

Tree size data from 2009 were used to select 12 *E. amplifolia* and 12 *E. grandis* representing small, medium, and large size classes (one whole tree and one tree for logs in each size class for each species for each season) in FORCE studies 102 and 102A (Tables 2, A1 and A2). The six *E. amplifolia* trees and six *E. grandis* trees for the spring field drying test were felled in February 2010, and the six *E. amplifolia* and six *E. grandis* in the summer test were felled in May 2010 (Table 3).

**Table 2.** Number and size ranges of *E. amplifolia* and *E. grandis* trees in the FORCE spring and summer studies and EH1 in the IRREC summer study.

| Study Season/Species | Number of Trees | Height (m) | DBH (cm) | Crown Length (m) | Crown Diameter (m) |
|---|---|---|---|---|---|
| FORCE Spring [1] | | | | | |
| *E. amplifolia* | 6 | 9.4–22.8 | 10.8–31.1 | 7.7–18.2 | 1.8–3.9 |
| *E. grandis* | 6 | 12.1–25.9 | 10.4–29.6 | 7.0–17.1 | 2.6–10.2 |
| FORCE Summer [2] | | | | | |
| *E. amplifolia* | 6 | 11.4–21.5 | 10.9–26.1 | 9.4–16.6 | 2.3–4.9 |
| *E. grandis* | 6 | 14.6–26.3 | 13.1–25.7 | 10.5–14.3 | 0.8–4.0 |
| IRREC Summer [3] | | | | | |
| EH1 | 19 | 9.5–18.9 | 4.1–12.9 | 2.8–8.5 | - |

[1] Individual trees listed in Table A1; [2] Individual trees listed in Table A2; [3] Individual trees listed in Table A3.

**Table 3.** Dates of monthly moisture content (MC$_{od}$) determinations of logs and trees in the FORCE and IRREC studies.

| Study Season | Months after Harvest | | |
|---|---|---|---|
| | 0 | 1 | 2 |
| FORCE Spring | 10 February 2010 | 10 March 2010 | 13 April 2010 |
| FORCE Summer | 12 May 2010 | 15 June 2010 | 13 July 2010 |
| IRREC Summer | 1 August 2022 | 2 September 2022 | 3 October 2022 |

In each season, two delimbed trees of each species were paired within three tree size classes (small, medium, and large). One tree in each pair was cut into 2.4 m logs; the other was kept whole. Logs and whole trees were stacked (slightly elevated on cottonwood (*P. deltoides* logs) to provide maximum drying and left out in the open field (Figure 1). Bark was intact on both logs and whole trees.

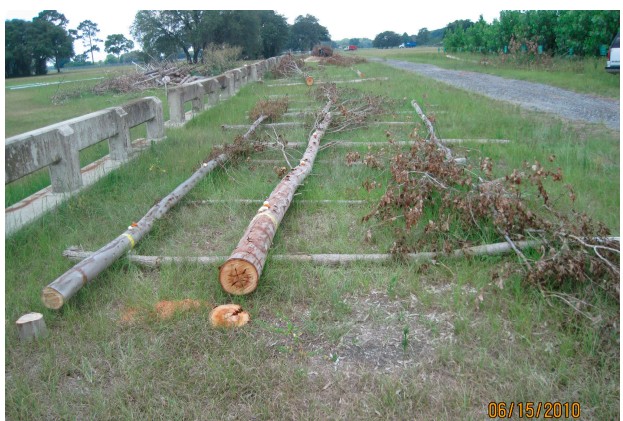 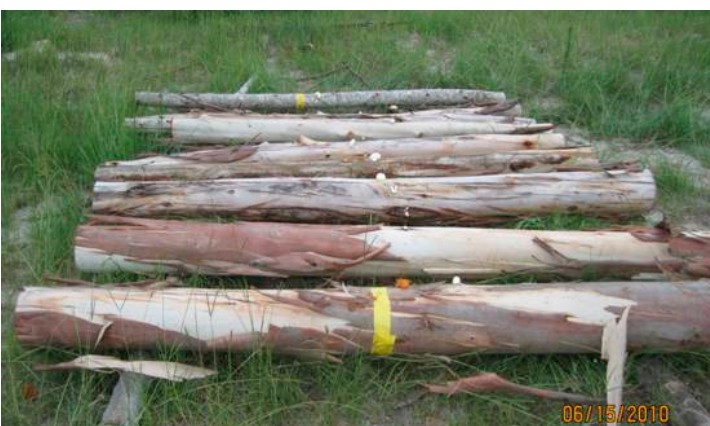

**Figure 1.** Stacked trees (**left**) and logs (**right**) in the FORCE Summer study.

In each season, stem cores were collected approximately monthly for two months after the season began. On each sampling date (Table 3), 12 mm diameter cores (bark-to-bark) were extracted from the middle of 2.4 m logs and corresponding stem positions (approximately 1.2, 3.6, 6, 8.5, 11, 13.5, 15.8, and 18.2 m heights = stem core positions 0, 1, 2, 3, 4, 5, 6, and 7, respectively) on whole trees using a powered increment borer. After extraction, the core holes were sealed immediately with expanding foam to avoid moisture loss from the holes. Each core was next weighed in the field and then dried in the lab at 101 °C until a constant weight was observed (Figure 2). After drying, core dry weights were taken, and $MC_{od}$ at the time of sampling was calculated on an oven dry weight (ODW) basis [29,30] as follows:

$$MC_{od} \text{ (\%)} = [(\text{Weight during sampling} - \text{ODW})/\text{ODW}] \times 100 \qquad (1)$$

Whole tree $MC_{od}$s were averages of component stem sections weighted by stem diameter.

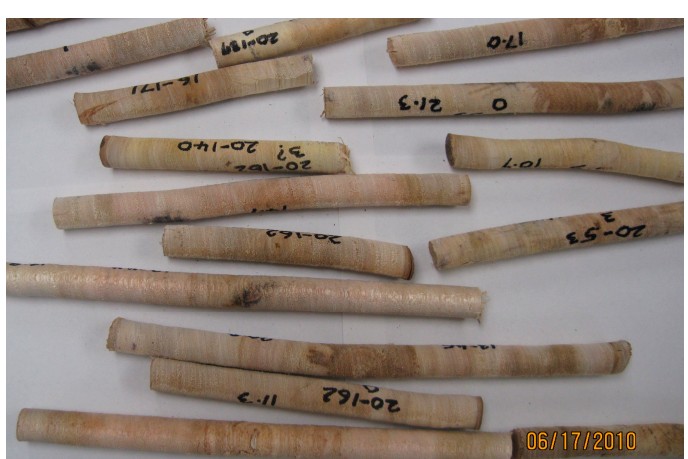 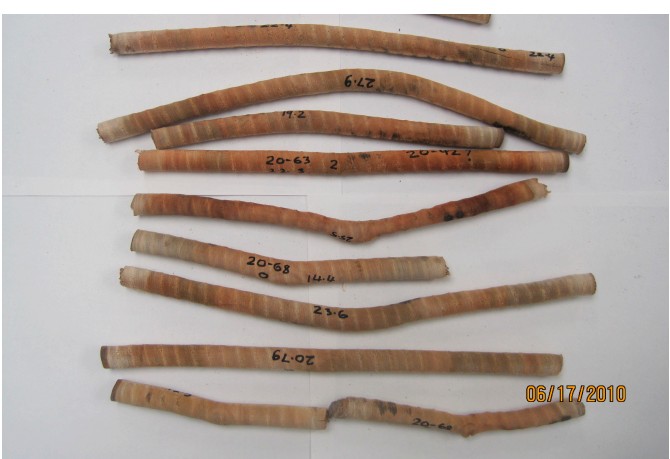

**Figure 2.** FORCE increment cores before (**left**) and after (**right**) oven drying at 101 °C.

Three *E. amplifolia* stem wood and bark samples weighing ~1 kg each were collected for chlorine analysis. These sample disks were bagged, refrigerated, and then shipped to

Standard Laboratories Inc. in Freeburg, Illinois, for chlorine content analysis using ASTM method D6721.

Statistical analyses of FORCE $MC_{od}$ data involved calculating standard errors for various means. These included stem cores at positions 0 to 7 by species, species–season, species–season–tree form–drying duration, and species–drying duration. Standard errors were also determined for whole tree means by species–season–drying duration–tree form.

### 2.2. IRREC Study

Cultivar EH1 was planted June 2015 on a sandy former pasture at the IRREC near Ft. Pierce, Florida (27°25′33″ N, 80°26′14″ W). Average maximum and minimum temperatures at IRREC are 27.7 and 18.3 °C, respectively, and average annual rainfall is 1368 mm. Trees in five three-row (twenty-six trees/row) plots received one of five fertilizers (control, GE 6-4-0 + micronutrients at 112, 224, and 336 kg of N ha$^{-1}$ rates, and diammonium phosphate equivalent to 336 kg of N ha$^{-1}$) and two replications of five-tree row plots of three planting densities (Table 1). This study was coppiced in June 2019. The interior row of each plot was periodically measured for tree size and number of coppice stems/stool at least half the DBH of the largest stem.

Tree size and planting density were used to select 18 EH1 sample trees representing the range of tree sizes (Tables 2 and A4). All 18 and an additional tree were felled in August 2022 (Table 3). Two trees in each of the three planting densities were paired for immediate sampling, one month of drying, and two months of drying. One tree in each 1-month and 2-month pairing was topped at the base of its crown; the other was kept whole. Nine topped and nine whole trees with bark were stacked on EH1 logs to promote drying. Seven ~13 cm long disks were taken near the 3.6 m stem position of the additional tree (from the 1667 planting density): two with bark were weighed fresh, five were subsequently cut into blocks without bark, and all seven were sun-dried for two months near Indiantown, Florida.

Due to the generally smaller sizes of the IRREC trees (Table 2), IRREC wood sampling was based on stem disks instead of cores. On each sampling date, ~10 cm long stem disks were cut at the 1.2, 3.6, 6, 8.5, and 11 m stem heights (disk positions 0, 1, 2, 3, and 4, respectively) of the 18 trees and weighed with bark. The bark was next removed from each disk of a tree, combined, and weighed fresh, as were individual disks, before drying at 101 °C until constant weight was observed. After drying, wood disk and whole tree bark dry weights were taken, and $MC_{od}$ was calculated on an ODW basis as in the FORCE study.

Statistical analyses of IRREC $MC_{od}$ data were limited to calculations of standard errors and means tests. Standard errors were determined for disk means by month-top-position combinations and for whole tree means by top-month combinations, top, and month. Means tests were performed for whole tree monthly $MC_{od}$ means.

Weather data for the two studies were obtained from the nearest UF/IFAS Florida Automated Weather Network (FAWN, https://fawn.ifas.ufl.edu accessed on 8 December 2022) weather stations: Okahumpka, FL, USA, for FORCE, and Ft. Pierce and Okeechobee, Florida, for IRREC. FAWN average daily temperature, relative humidity, and evapotranspiration (ET) data were averaged and summed, respectively, for the monthly and bimonthly periods in the two studies (Table 4).

Drying conditions were lower for the FORCE spring season and similar for the FORCE and IRREC summer seasons (Table 4). The first month of the FORCE spring season had less than half of the ET of the summer months and combined with the second FORCE spring month, the FORCE spring season had 8–14 cm less ET than the summer seasons. The IRREC summer at Ft. Pierce was similar to the IRREC season experienced by the EH1 blocks dried at Indiantown.

**Table 4.** Average temperatures (T, °C), relative humidities (RH, %), and total evapotranspiration (ET, cm) during drying periods in the FORCE and IRREC studies.

| Variable | Months after Harvest | | |
|---|---|---|---|
| | 0–1 | 1–2 | 0–2 |
| | FORCE Spring | | |
| T; RH; ET | 10.1; 67; 5.59 | 18.1; 73; 10.08 | 14.4; 71; 15.67 |
| | FORCE Summer | | |
| T; RH; ET | 26.0; 75; 16.23 | 27.3; 82; 13.11 | 26.8; 78; 29.34 |
| | IRREC Summer (Ft. Pierce) | | |
| T; RH; ET | 27.8; 75; 14.61 | 25.9; 77; 10.52 | 26.8; 76; 25.12 |
| | IRREC Summer (Okeechobee) | | |
| T; RH; ET | 26.9; 84; 13.03 | 25.4; 87; 10.13 | 26.1; 86; 23.16 |

## 3. Results and Discussion

### 3.1. FORCE Study

The average initial $MC_{od}$ ranged between 136 and 96.2% at stem core positions 0 and 7, respectively, in *E. grandis* and from 120 and 112.3% at *E. amplifolia* stem core positions 5 and 4. Initial $MC_{od}$ declined with height in *E. grandis* but was nearly constant in *E. amplifolia* (Figure 3, Table A4).

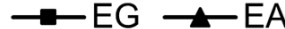

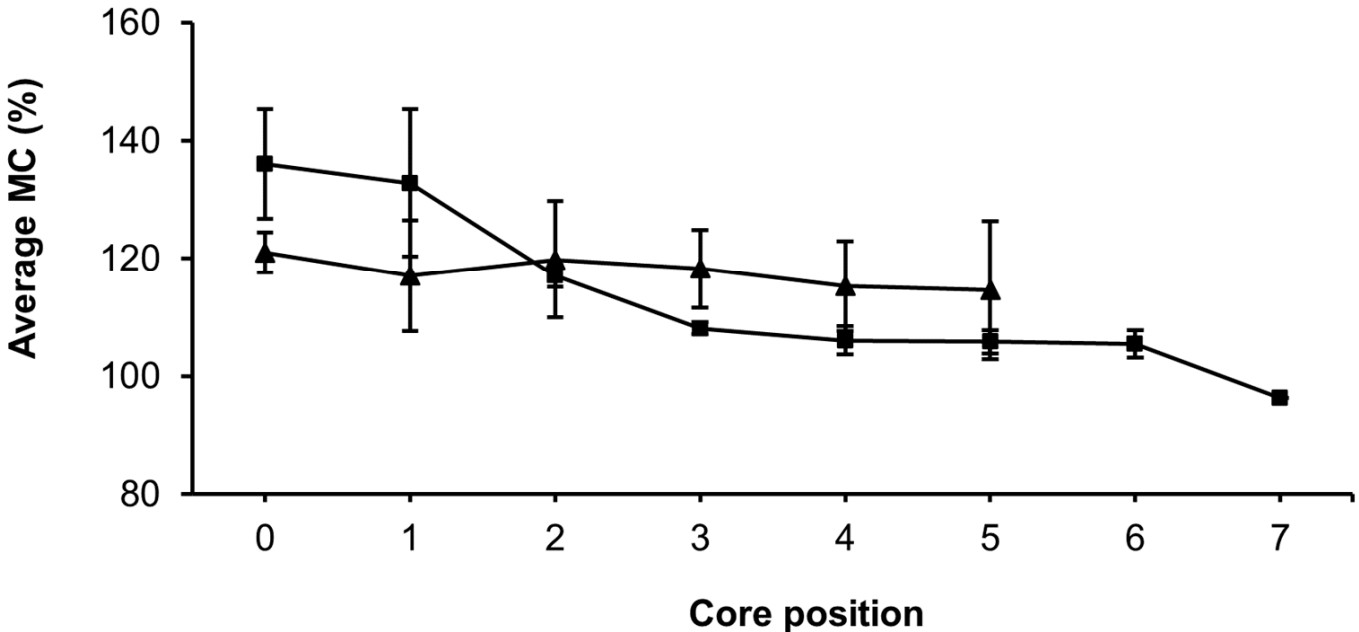

**Figure 3.** Average initial moisture content ($MC_{od}$) in *E. grandis* (EG) and *E. amplifolia* (EA) trees at stem core positions 0 to 7 in the FORCE study.

The average initial $MC_{od}$ was similar in the spring and summer seasons for *E. grandis* and *E. amplifolia* trees (Figure 4, Table A4). Trees generally showed a slight decline in $MC_{od}$ with height except in the spring *E. amplifolia* trees. Combining both tree forms, *E. grandis* whole tree $MC_{od}$ was 122% in spring and 114% in the summer, compared with *E. amplifolia*'s 119% in spring and 117% in summer.

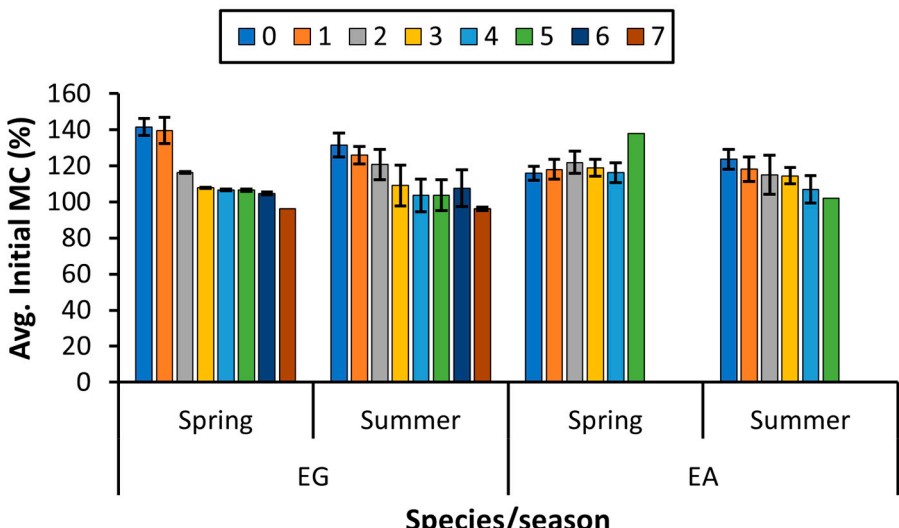

**Figure 4.** Average initial moisture content (MC$_{od}$) at *E. grandis* (EG) and *E. amplifolia* (EA) stem core positions 0 to 7 in the spring and summer seasons in the FORCE study.

An earlier Florida study [9] also noted decreasing *E. grandis* MC with height. Decreasing MC with height in *Eucalyptus*, however, contrasts with the findings in *Pinus*. Clark and Daniels [31] observed that MC varied in three age groups and increased with height in loblolly pine (*P. taeda* L.). Stohr [32] also reported increasing MC with height in loblolly pine, slash pine (*P. elliottii* Engelm.), and Mexican weeping pine (*P. patula* Schiede ex Schlechtendal et Chamisso), with a fairly constant MC after 50% height in Mexican weeping pine. McMinn and Taras [9] also reported increasing MC with height in slash pine. In Western hemlock (*Tsuga heterophylla* (Raf.) Sarg.), a relatively constant MC was observed throughout the stem [33]. The highest MC$_{od}$ (120%) at position 5 in EA was probably due to the single sample point at that height.

The average initial whole tree MC$_{od}$ (i.e., at 0 months) was similar in the spring and summer seasons for *E. grandis* and *E. amplifolia* trees (Figure 5, Table A4). *E. grandis* whole tree MC$_{od}$ was 122% in spring and 116% in the summer, compared with *E. amplifolia*'s 119% in spring and 117% in summer. Similar observations have been made in softwoods such as loblolly pine and slash pine [34], but significant differences have been observed between species, geographic location, and the interaction of the two. However, Patterson and Doruska [35] reported seasonal MC variation in loblolly pine.

Both *E. grandis* and *E. amplifolia* lost moisture at a higher rate in the first month (Table 5, Figure 6) which is consistent with the observations in other studies [29,30]. The average MC$_{od}$ in *E. grandis* ranged between 63.5 and 86.1% in the first month and between 70.7 and 102.1% in *E. amplifolia*. Moisture loss has been observed as early as a week after felling hardwoods such as red oaks (*Quercus* spp.), sweetgum (*Liquidambar styraciflua* L.), and yellow poplar (*Liriodendron tulipifera* L.) [8]. Most of the free water is removed during this initial drying period due to the weak capillary force holding it [30,36]. *Eucalyptus* wood tends to have relatively more free water. Free water starts moving out of the wood in the form of drips from cut ends as soon as the trees are cut or via evapotranspiration by trees with intact foliage. The moisture gradient of the wood and its environment continues to decrease until the wood moisture is approximately 25–30%, at which point FSP is reached. Any additional moisture removal below this level will require additional energy due to the stronger bonds that hold bound water. In the second month, the average MC$_{od}$ in *E. grandis* ranged between 57.8 and 71.1% and between 54.4 and 82.9% in *E. amplifolia*.

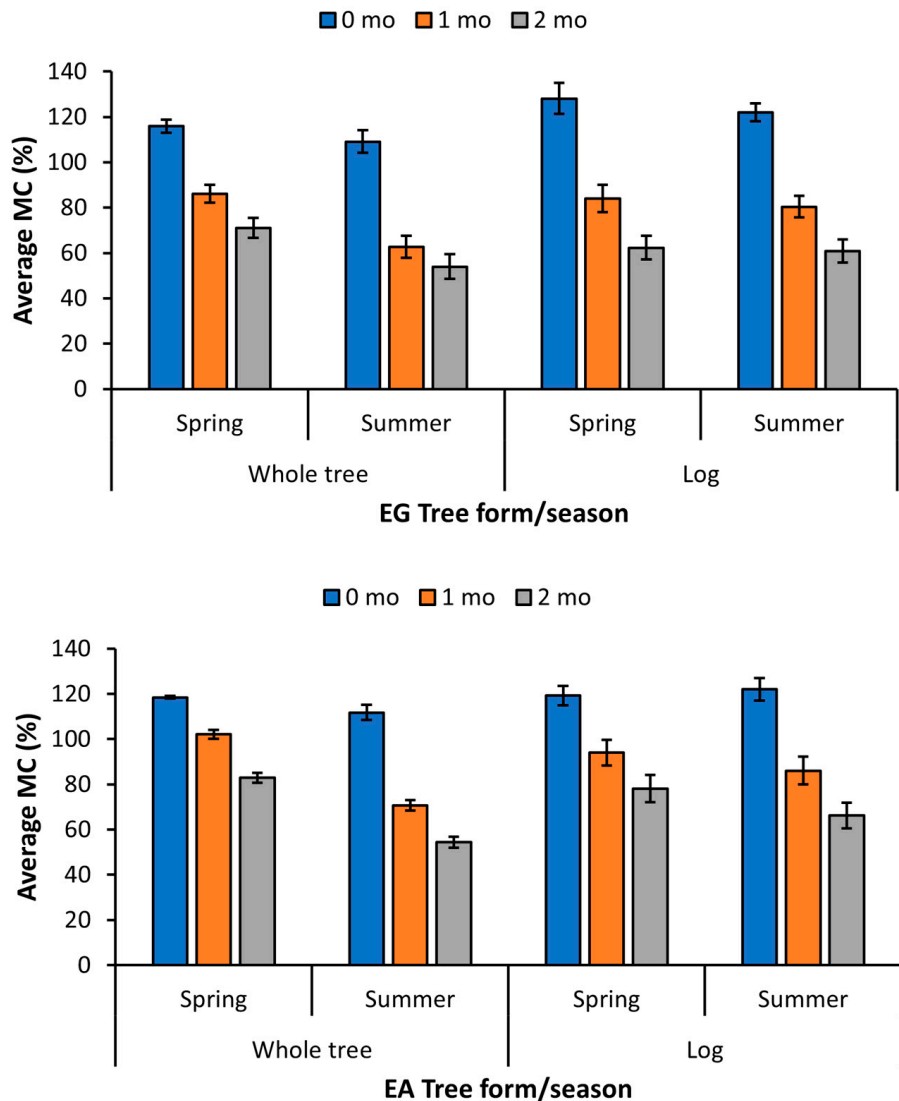

**Figure 5.** *E. grandis* (EG, **top**) and *E. amplifolia* (EA, **bottom**) whole tree average moisture content (MC$_{od}$) at three times after harvest (0, 1, or 2 months) by tree form (whole tree or log) and season (Spring or Summer).

**Table 5.** Number of trees (*n*) and whole tree average moisture content (MC$_{od}$, %) by species, tree form, season, and drying duration (months) in the FORCE and IRREC studies.

| Study-Species | Form | Season | *n* | MC by Duration | | |
|---|---|---|---|---|---|---|
| | | | | 0 | ~1 | ~2 |
| FORCE-*E. grandis* | Whole | Spring | 3 | 115.9 | 86.1 | 71.1 |
| | | Summer | 3 | 106.0 | 63.5 | 57.8 |
| | Log | Spring | 3 | 128.2 | 83.7 | 62.4 |
| | | Summer | 3 | 122.0 | 80.4 | 60.9 |
| FORCE-*E. amplifolia* | Whole | Spring | 3 | 118.5 | 102.1 | 82.9 |
| | | Summer | 3 | 111.8 | 70.7 | 54.4 |
| | Log | Spring | 3 | 119.2 | 94.0 | 78.1 |
| | | Summer | 3 | 122.1 | 86.1 | 66.2 |
| IRREC-EH1 | No Top | Summer | 3 | | 38.3 | 60.1 |
| | Top | Summer | 6; 3; 3 | 130.5 | 47.9 | 46.2 |
| | Blocks | Summer | 7 | | - | 19.4 |

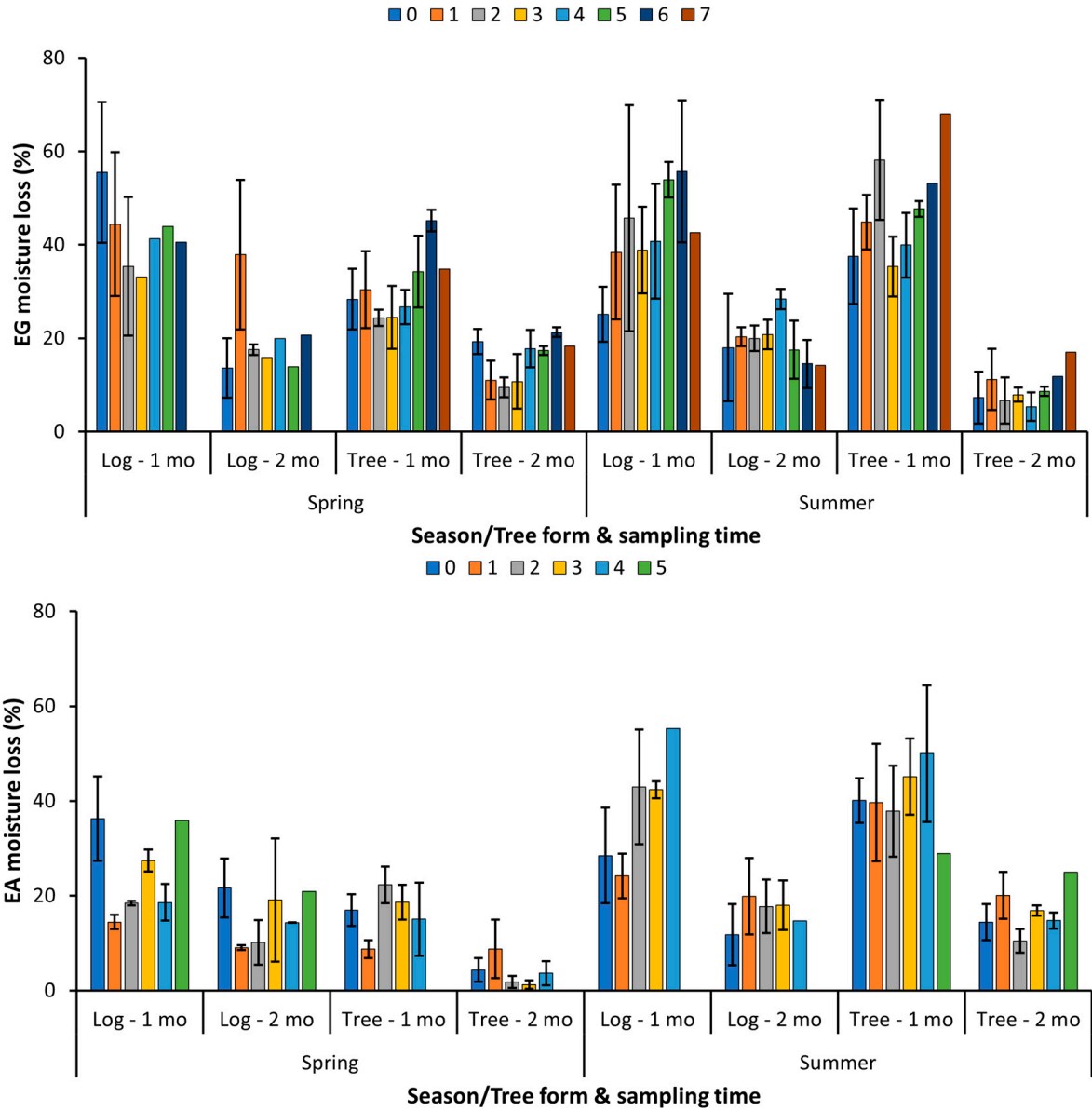

**Figure 6.** *E. grandis* (EG, **top**) and *E. amplifolia* (EA, **bottom**) moisture content (MC_od) loss from stem core positions 0 to 7 by season (Spring or Summer), tree form (whole tree or log), and time after harvest (1 or 2 months).

Summer season drying tended to exceed spring drying, both for trees and logs (Table 5). For example, *E. grandis* trees after one month averaged 63.5% in the summer compared with 86.1% in the spring, while *E. grandis* logs averaged 80.4% in the summer and 83.7% in the spring. *E. amplifolia* and *E. grandis* whole trees dried more than logs in the summer (Table 5). After one month, trees were over 15% drier than logs.

In the first month, the rate of moisture loss was higher towards the top in *E. grandis* but remained fairly constant throughout the tree in *E. amplifolia*. The highest moisture loss of 49.3% was observed at stem core position 6 in *E. grandis*, but the lowest loss of 31.8% was at position 3. The extent of moisture loss in *E. grandis* from the higher heights was similar to the observations made by McMinn and Taras [9] in the same species—50% moisture loss in one month. In *E. amplifolia*, the highest loss of 33.4% was observed at position 3. Higher moisture loss from small diameter stems or smaller wood samples is expected because water has to travel less distance from inside the wood to the surface [26,29,30]. In addition, tree tops are dominated by sapwood resulting in higher water storage but faster water

flow [30,37,38]. Moisture loss in the second month was fairly constant and similar in both *E. grandis* and *E. amplifolia*.

Consequently, drying duration had slightly different effects in *E. amplifolia* and *E. grandis*. The absolute $MC_{od}$ in *E. grandis* was less than 100% at all stem positions after 1 month and less than 80% after 2 months of drying (Figure A1). While the absolute $MC_{od}$s in *E. amplifolia* were similar after 1 and 2 months of drying, and the $MC_{od}$s were very uniform up the stem (Figure A2).

Other studies suggest that season significantly impacts the rate of drying, with more drying possible if trees are left in the field longer. Within two weeks of harvest, slash pine MC was reduced by 1–18% in winter but 14–41% in summer [9]. Transpirational drying reduced whole tree MC of the sitka spruce (*Picea sitchensis* (Bong.) Carr.) by 13% over a 5-month period while the reduction was only 17% in lodgepole pine (*P. contorta* Dougl.) over one winter and two summers [39].

The chlorine content in *E. amplifolia* wood ranged from 0.15 to 0.49% (Table 6). These values are higher than the 0.07% previously reported for *E. grandis* grown in central Florida [40] and suggest that using *E. amplifolia* in certain biomass energy plants may be problematic.

**Table 6.** Chlorine content in *E. amplifolia* wood in the FORCE study.

| Tree ID | Stem Height (m) | Chlorine Content (%) |
|---|---|---|
| 16-78E | 2.4 | 0.15 |
| 16-70E | 4.8 | 0.32 |
| 20-68 | 4.8 | 0.49 |

*3.2. IRREC Study*

The average initial $MC_{od}$ for EH1 coppice ranged between 138.0 and 114.2% at stem disk positions 0 and 4, respectively (Table 7). Initial MC declined over 20% with height, a pattern similar to that of *E. grandis* in the FORCE study (Figure 3).

**Table 7.** EH1 stem disk moisture content ($MC_{od}$, %) means and standard errors for disk position by month and top in the IRREC study.

| Month-Top | *n* | MC by Disk Position | | | | | |
|---|---|---|---|---|---|---|---|
| | | 0 | 1 | 2 | 3 | 4 | Average |
| 0-All | 6 | 138.0 ± 3.2 | 135.0 ± 4.2 | 127.3 ± 3.6 | 121.1 ± 4.4 | 114.2 ± 3.5 | 130.5 |
| 1-All | 6 | 54.4 ± 9.9 | 42.7 ± 9.5 | 34.7 ± 7.6 | 33.5 ± 7.8 | 32.9 ± 3.7 | 43.1 |
| 2-All | 6 | 69.6 ± 4.3 | 56.3 ± 5.2 | 41.9 ± 3.5 | 36.4 ± 2.9 | 34.8 ± 5.0 | 53.2 |
| 1-No Top | 3 | 47.7 ± 18.4 | 37.9 ± 19.7 | 28.1 ± 14.6 | 37.1 ± 18.7 | 22.8 | 38.3 |
| 1-Top | 3 | 61.0 ± 10.4 | 47.6 ± 6.1 | 41.2 ± 6.0 | 31.0 ± 7.7 | 37.9 ± 2.3 | 47.9 |
| 2-No Top | 3 | 75.5 ± 6.5 | 65.5 ± 7.0 | 48.1 ± 4.6 | 40.0 ± 4.9 | 45.3 ± 3.1 | 60.1 |
| 2-Top | 3 | 63.8 ± 3.9 | 47.1 ± 1.0 | 35.7 ± 0.6 | 32.8 ± 2.4 | 24.2 ± 1.8 | 46.2 |

A similar pattern was observed in the initial $MC_{od}$ of stem disks from one of two larger *E. urophylla* clones in Brazil [41]. One clone, 20.3 cm in DBH, had basal, midstem, and upperstem MCs of 121.5, 112.6, and 126.2%, respectively, while the 19.4 cm DBH clone had corresponding MCs of 94.5, 84.5, and 81.0%.

After two months of field drying, EH1 $MC_{od}$ at all disk positions had decreased by more than 70%, averaging 53.2% across all five positions with and without tops (Table 8). EH1 coppice with retained tops had 8 to 21% less $MC_{od}$, depending on disk position, with an average decrease of nearly 14%. Whole tree $MC_{od}$ decreased with the duration of field drying and was lowest at 46.2% after two months of drying trees with tops (Tables 8 and A4).

**Table 8.** EH1 whole tree moisture content ($MC_{od}$, %) means and standard errors by top and month of drying factors in the IRREC study.

| Factor | | MC Average | |
| --- | --- | --- | --- |
| Top | Month | Top | Month [1] |
| Yes | 0 | 130.5 ± 3.1 | 130.5 ± 3.1 A |
| No | | 38.3 ± 12.7 | |
| Yes | 1 | 47.9 ± 5.8 | 43.1 ± 7.3 B |
| No | | 60.1 ± 3.9 | 53.2 ± 3.5 B |
| Yes | 2 | 46.2 ± 1.1 | |

[1] Letters indicate differences among means at the 5% level.

Field drying stem disks and blocks appears to lower $MC_{od}$ even more. Averaging 19.4% MC after two months of drying at Indiantown (Table 5), the two wood disks with bark intact and five squared blocks without bark suggest that cutting and piling short stem sections in open air piles has merit.

Using large field dried trees directly for feedstock in energy facilities can be challenging because of inconsistent fuel moisture across stem sections as the diameter influences the drying rate. Moreover, a high MC reduces the efficiency of the energy facilities. The results of this study and others [30,38] show small diameter stems can dry faster and have relatively consistent moisture. Because of the higher productivity, adaptation to various climates, and its ability to coppice, *Eucalyptus* therefore has the potential to supply year-round quality feedstock. The field drying of *E. saligna* Sm. SRWC coppice was an essential component of year-round harvesting to fuel a proposed 10MW bioenergy plant in New Zealand [42]. Based on field trials of 3- and 5-year-old trees cut throughout the year that showed that the MC initially dropped rapidly in the first few weeks after cutting regardless of season and then declined at various rates [43], harvesting would be continual throughout the year, and the biomass would be stored on the farms or at the plant.

Biochar pyrolysis plants, such as Green Carbon Solutions' (GCS) [22], are designed to be continuous, to minimize electricity use, and to capture and convert volatiles into usable forms of energy. A wood supply with a low MC will contribute to cost-effectively producing biochar for use in applications such as soil nutrient and water retention, the remediation of contaminated soils and water, filler in various products, acoustic and thermal insulation, carbon fibers and polymers, filtration media, carbon sequestration, and heavy metal adsorption [22]. As producers look to increase the quality of biochar, so the woody biomass quality and associated specifications become increasingly important. Beyond biomass specific specifications, a low and consistent MC is a key contributor to enhancing biochar consistency and quality.

Though field drying can significantly reduce wood MC, there are still some logistical issues that need to be addressed. In traditional uses, wood is transported to the facility soon after harvest and dried. *Eucalyptus* wood contains relatively more moisture, sometimes more than the wood itself. If drying is performed at the facility, a lot of money will have to be spent to transport the water. Field drying seems ideal in that regard, but equipment may not be available on site on later dates for loading wood as equipment is usually moved to other sites after harvest. As is often the case, producers will need to evaluate economic tradeoffs, and in this case, the pros and cons of field drying versus paying for transport of water.

## 4. Conclusions

These studies assessed the potential effectiveness of field drying *Eucalyptus* logs and trees for twelve combinations of species, seasons, drying durations, and tree sizes and forms. Species, tree size, and tree form were not important in reducing wood $MC_{od}$, but harvest season and drying duration affected $MC_{od}$, which decreased with longer storage and increased seasonal temperature, relative humidity, and evapotranspiration. The results of the FORCE and IRREC studies suggest that field drying can improve wood quality

for biochar and other products by reducing wood $MC_{od}$, perhaps most when short stem sections are dried. However, the higher chlorine content in *E. amplifolia* wood may be a concern for its use in some technologies.

**Author Contributions:** Conceptualization, D.L.R., B.T. and M.F.E.; Methodology, D.L.R., B.T. and M.F.E.; Software, D.L.R. and B.T.; Validation, D.L.R., B.T. and M.F.E.; Formal Analysis, D.L.R. and B.T.; Investigation, D.L.R., B.T. and M.F.E.; Resources, D.L.R., B.T. and M.F.E.; Data Curation, D.L.R. and B.T.; Writing—Original Draft Preparation, D.L.R. and B.T.; Writing—Review and Editing, D.L.R., B.T. and M.F.E.; Visualization, D.L.R.; Supervision, D.L.R.; Project Administration, D.L.R.; Funding Acquisition, D.L.R. and M.F.E. All authors have read and agreed to the published version of the manuscript.

**Funding:** This research received no external funding, and the APC was funded by Florida FGT.

**Data Availability Statement:** Whole tree data for these studies are provided in Appendix A. Core and stem position data are available upon request to the senior author.

**Acknowledgments:** The authors gratefully acknowledge the direct and/or indirect support provided by Buckeye Technologies, Inc., ADAGE (an AREVA and Duke Energy joint venture), FORCE, Sumter County, the IRREC, GCS, and ArborGen.

**Conflicts of Interest:** The authors declare no conflict of interest.

## Appendix A

**Table A1.** Description of six *E. amplifolia* (EA) and six *E. grandis* (EG) trees in the FORCE spring season study.

| Tree ID | Species | Tree Form | Class Size | Height (m) | DBH (cm) | Crown Length (m) | Crown Width (m) |
|---------|---------|-----------|------------|------------|----------|------------------|-----------------|
| 16-78E [a] | EA | Tree | Large | 19.5 | 31.1 | 16.2 | 4.0 |
| 16-70E [a] | EA | Log | Large | 22 | 26.1 | 15.8 | 3.4 |
| 20-68 [a] | EA | Log | Medium | 22.8 | 21.2 | 17.4 | 3.9 |
| 20-63 | EA | Tree | Medium | 20.5 | 20.0 | 18.2 | 4.7 |
| 20-40 | EA | Tree | Small | 9.4 | 11.5 | 7.4 | 1.8 |
| 20-28 | EA | Log | Small | 12.8 | 10.8 | 11.4 | 2.3 |
| 20-42 | EG | Tree | Large | 25.9 | 29.6 | 15.0 | 7.3 |
| 20-104 | EG | Tree | Large | 23.8 | 27.5 | 17.1 | 10.2 |
| 20-58 | EG | Log | Medium | 24.5 | 21.2 | 17.1 | 6.0 |
| 20-47 | EG | Tree | Medium | 21.3 | 21.1 | 17.1 | 6.0 |
| 20-100 | EG | Log | Small | 13.1 | 19.2 | 9.0 | 2.6 |
| 20-41 | EG | Log | Small | 12.1 | 10.4 | 7.0 | 3.3 |

[a] Tree used for chlorine analysis.

**Table A2.** Description of six *E. amplifolia* (EA) and six *E. grandis* (EG) trees in the FORCE summer season study.

| Tree ID | Species | Tree Form | Class Size | Height (m) | DBH (cm) | Crown Length (m) | Crown Width (m) |
|---------|---------|-----------|------------|------------|----------|------------------|-----------------|
| 20-23 | EA | Log | Large | 21.5 | 26.1 | 15.9 | 4.9 |
| 20-137 | EA | Tree | Large | 20.0 | 25.2 | 16.5 | 4.4 |
| 20-140 | EA | Tree | Medium | 18.3 | 21.5 | 16.6 | 3.8 |
| 20-25 | EA | Log | Medium | 19.2 | 21.4 | 16.5 | 4.2 |
| 20-139 | EA | Tree | Small | 12.5 | 11.5 | 10.0 | 2.3 |
| 16-40E | EA | Log | Small | 11.4 | 10.9 | 9.4 | 2.9 |
| 13-44 | EG | Tree | Large | 26.3 | 25.7 | 12.9 | 2.8 |
| 13-45 | EG | Log | Large | 24.3 | 25.5 | 13.4 | 2.9 |
| 20-53 | EG | Log | Medium | 25.5 | 22.3 | 14.3 | 3.7 |
| 16-171E | EG | Tree | Medium | 21.2 | 21.6 | 12.2 | 4.0 |
| 20-113 | EG | Log | Small | 19.5 | 13.9 | 11.6 | 1.8 |
| 20-162 | EG | Tree | Small | 14.6 | 13.1 | 10.5 | 0.8 |

**Table A3.** Planting density (trees ha$^{-1}$), top retained, months of drying, DBH (cm), height (m), crown height (m), and whole tree moisture content (MC$_{od}$, %) of 19 EH1 trees in the IRREC study.

| Tree ID | Density | Top | Month | DBH | Height | Crown Height | MC |
|---|---|---|---|---|---|---|---|
| 1 | 3333 | Yes | 0 | 8.7 | 15.5 | 11.9 | 136.6 |
| 2 | 3333 | No | 2 | 6.1 | 13.4 | 9.5 | 50.8 |
| 3 | 3333 | No | 1 | 4.1 | 10.1 | 6.4 | 22.4 |
| 4 | 3333 | Yes | 0 | 7.3 | 14.3 | 9.8 | 138.5 |
| 5 | 3333 | Yes | 2 | 8.8 | 18.9 | 11.9 | 46.9 |
| 6 | 3333 | Yes | 1 | 5.1 | 11.9 | 9.1 | 34.1 |
| 7 | 1111 | Yes | 0 | 12.9 | 18.3 | 9.8 | 123.9 |
| 8 | 1111 | No | 2 | 9.9 | 17.4 | 9.8 | 63.1 |
| 9 | 1111 | No | 1 | 4.9 | 9.5 | 4.3 | 23.0 |
| 10 | 1111 | Yes | 2 | 10.7 | 17.7 | 11.0 | 48.0 |
| 11 | 1111 | Yes | 1 | 9.0 | 16.5 | 10.7 | 52.0 |
| 12 | 1111 | Yes | 0 | 9.4 | 16.8 | 10.4 | 138.3 |
| 13 | 1667 | No | 1 | 8.7 | 15.3 | 9.5 | 69.5 |
| 14 | 1667 | No | 2 | 11.8 | 17.7 | 9.5 | 66.4 |
| 15 | 1667 | Yes | 0 | 8.8 | 17.1 | 9.5 | 119.4 |
| 16 | 1667 | Yes | 1 | 10.3 | 17.4 | 9.5 | 57.6 |
| 17 | 1667 | Yes | 2 | 10.8 | 15.5 | 9.8 | 43.7 |
| 18 | 1667 | Yes | 0 | 10.9 | 18.0 | 9.8 | 126.4 |
| G | 1667 | No | 2 | 13.4 | - | - | |

**Table A4.** Moisture content (MC$_{od}$, %)) by stem core position for species (*E. grandis* (EG), *E. amplifolia* (EA)), tree form, season, and approximate drying duration (months) combinations in the FORCE study.

| Species | Form | Season | Duration | MC by Core Position | | | | | | | |
|---|---|---|---|---|---|---|---|---|---|---|---|
| | | | | 0 | 1 | 2 | 3 | 4 | 5 | 6 | 7 |
| EG | Whole | Spring | 0 | 136.0 | 132.8 | 117.0 | 108.1 | 106.0 | 105.8 | 105.4 | 96.2 |
| | | | 1 | 107.7 | 102.4 | 92.6 | 83.6 | 79.3 | 71.6 | 60.2 | 61.4 |
| | | | 2 | 88.4 | 91.4 | 83.2 | 72.9 | 61.6 | 54.2 | 38.9 | 43.1 |
| | | Summer | 0 | 127.6 | 119.9 | 102.8 | 94.8 | 91.1 | 91.7 | 98.3 | 97.2 |
| | | | 1 | 96.8 | 78.3 | 57.1 | 59.4 | 51.2 | 44.1 | 45.2 | 29.1 |
| | | | 2 | 95.0 | 73.6 | 55.3 | 51.5 | 45.8 | 35.4 | 33.3 | 12.2 |
| | Log | Spring | 0 | 147.2 | 146.5 | 115.2 | 107.0 | 108.4 | 107.8 | 103.1 | - |
| | | | 1 | 91.7 | 102.1 | 79.8 | 73.8 | 63.1 | 63.8 | 62.5 | - |
| | | | 2 | 78.0 | 64.2 | 62.3 | 58.0 | 47.2 | 50 | 41.8 | - |
| | | Summer | 0 | 131.7 | 128.6 | 128.2 | 123.3 | 116.0 | 115.8 | 112.4 | 95.1 |
| | | | 1 | 106.6 | 90.2 | 82.5 | 84.4 | 75.3 | 61.9 | 56.6 | 52.5 |
| | | | 2 | 88.6 | 69.9 | 62.6 | 63.6 | 47.0 | 44.4 | 42.1 | 38.3 |
| EA | Whole | Spring | 0 | 121.0 | 117.0 | 119.8 | 118.2 | 115.3 | - | - | - |
| | | | 1 | 104.0 | 108.3 | 97.5 | 99.6 | 100.3 | - | - | - |
| | | | 2 | 86.2 | 85.5 | 84.1 | 79.5 | 77.4 | - | - | - |
| | | Summer | 0 | 116.7 | 115.2 | 107.2 | 111.9 | 108.9 | - | - | - |
| | | | 1 | 76.6 | 75.5 | 69.4 | 66.7 | 58.9 | - | - | - |
| | | | 2 | 62.2 | 55.4 | 58.9 | 49.9 | 44.2 | - | - | - |
| | Log | Spring | 0 | 110.9 | 119.1 | 124.0 | 119.7 | 117.2 | 138.1 | - | - |
| | | | 1 | 74.6 | 104.7 | 105.6 | 92.3 | 98.6 | 102.2 | - | - |
| | | | 2 | 53.0 | 95.6 | 95.4 | 73.2 | 84.3 | 81.3 | - | - |
| | | Summer | 0 | 130.7 | 122.5 | 123.3 | 117.0 | 103.3 | 102.0 | - | - |
| | | | 1 | 102.2 | 98.3 | 80.1 | 74.7 | 48.0 | 73.1 | - | - |
| | | | 2 | 77.9 | 78.4 | 62.3 | 56.6 | 33.3 | 48.1 | - | - |

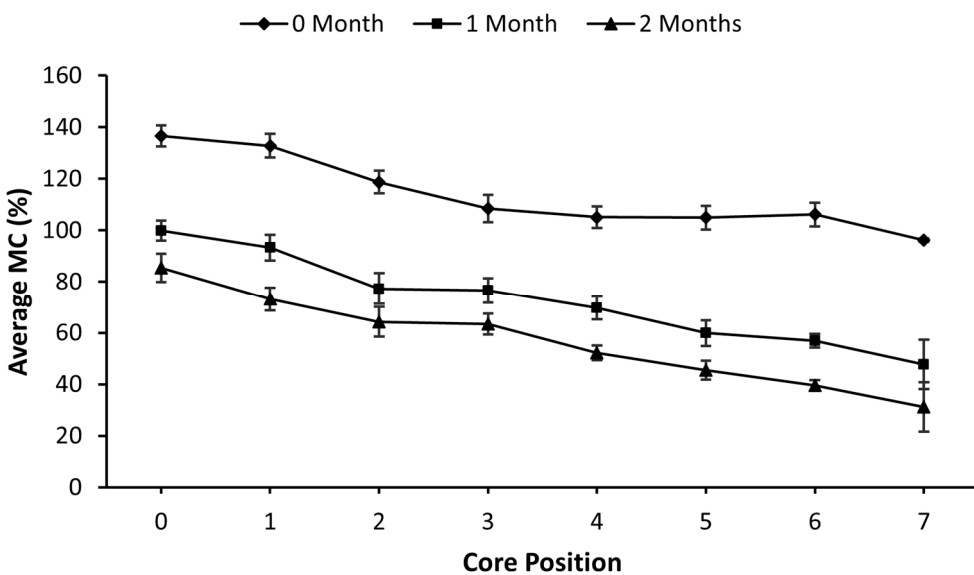

**Figure A1.** *E. grandis* moisture content (MC) by stem core position and drying duration in the FORCE study.

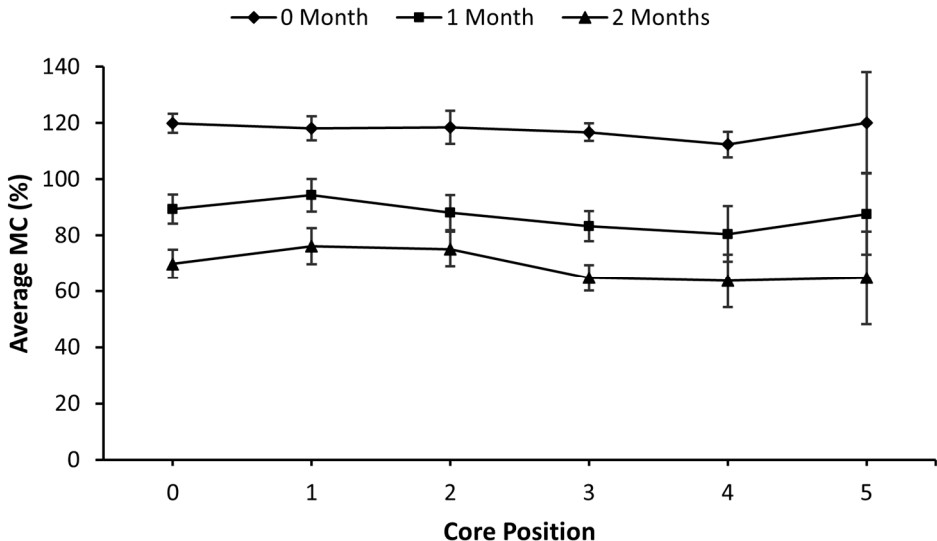

**Figure A2.** *E. amplifolia* moisture content (MC) by stem core position and drying duration in the FORCE study.

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
