# Peer review of "Field Drying for Enhancing Biomass Quality of Eucalyptus Logs and Trees in Florida, USA"

_forests, doi:10.3390/f14050899_

Round 1
Reviewer 1 Report (Previous Reviewer 1)
Accept
Author Response
Reviewer 1's comment of Accept is appreciated.
Reviewer 2 Report (Previous Reviewer 2)
The authors have introduced the necessary corrections. Please note the comment:
Abstract:
Line 12: … 24 6- and …
- What does this record mean? There was a mistake in the record here.
Author Response
Addressing Reviewer 2's suggestion about:
Line 12: "… 24 6- and …"
What does this record mean? There was a mistake in the record here.
this has been reworded to "...12 6-year-old and 12 8-year-old..."
Reviewer 3 Report (Previous Reviewer 4)
The article discusses the evaluation of the process of natural drying of Eucalyptus wood. amplifolia and E. grandis E. urophylla. The authors presented assumptions on the evaluation of the process of natural drying of wood of selected species. Significantly atmospheric conditions on the change of final moisture content were indicated.
Wood harvested from fast-growing plantations was presented as a source of fuels. The necessity of drying wood to the appropriate moisture content was indicated.
The characteristics of the research raw material were correctly presented. Growing conditions and water and soil requirements were given.
Comments:
The reason for the difference in the sampling of wood from the FORCE group and the IRREC group was imprecisely explained.
The reason for the change in drying methods was imprecisely stated - was part of the log debarked?
For whole trees, the research process is comparable. However, why did the split logs from the groups have different lengths?
The discussion lacked an explanation of how the transpiration process works to decrease moisture content in whole trees.
There was no explanation of how strong was the gradient of moisture change resulting from the drying rate and the difference in equilibrium humidity of the air. It would be worth expanding the paper to indicate the effect of ambient conditions on moisture changes in stored wood
There was no indication of how the drying process was affected by increased cross-sectional or longitudinal area. The cross-sectional area is particularly susceptible to water diffusion (shortening logs increases the face area relative to the volume of roundwood).
Conclusions seem too laconic. There is no reference to the differences between methods of drying wood and what is the influence of external conditions (hygroscopic equilibrium - ambient temperature and humidity, as a parameter affecting psychrometric differences). It is worth referring in the conclusions to the indication of the value of the decrease in moisture content depending on the period of storage and temperature and humidity conditions of the air.
The paper lacked an analysis of the results confirming whether the obtained results are statistically significant.
Author Response
Reviewer 3's nine Comments were addressed as indicated after the CAPS:
1. The reason for the difference in the sampling of wood from the FORCE group and the IRREC group was imprecisely explained. ADDED "Due to the generally smaller sizes of the IRREC trees (Table 2), IRREC wood sampling was based on stem disks instead of cores." to the IRREC Materials and Methods.
2. The reason for the change in drying methods was imprecisely stated - was part of the log debarked? ADDED "Nine topped and nine whole trees with bark..." to indicate that bark was not removed from 18 IRREC trees.
3. For whole trees, the research process is comparable. However, why did the split logs from the groups have different lengths? NO CHANGE as the FORCE stem cores and IRREC stem disks were taken at the same positions at 2.4m lengths.
4. The discussion lacked an explanation of how the transpiration process works to decrease moisture content in whole trees. ADDED "...or via evapotranspiration by trees with intact foliage."
5. There was no explanation of how strong was the gradient of moisture change resulting from the drying rate and the difference in equilibrium humidity of the air. It would be worth expanding the paper to indicate the effect of ambient conditions on moisture changes in stored wood. NO CHANGE as this aspect is outside the scope of our ms.
6. There was no indication of how the drying process was affected by increased cross-sectional or longitudinal area. The cross-sectional area is particularly susceptible to water diffusion (shortening logs increases the face area relative to the volume of roundwood). NO CHANGE as this aspect is also outside the scope of our ms.
7. Conclusions seem too laconic. There is no reference to the differences between methods of drying wood and what is the influence of external conditions (hygroscopic equilibrium - ambient temperature and humidity, as a parameter affecting psychrometric differences). It is worth referring in the conclusions to the indication of the value of the decrease in moisture content depending on the period of storage and temperature and humidity conditions of the air. ADDED "..., which decreased with longer storage and increased seasonal temperature, relative humidity, and evapotranspiration."
9, The paper lacked an analysis of the results confirming whether the obtained results are statistically significant. NO CHANGE as the statistics were deemphasized based on a previous reviewer's suggestion.
Reviewer 4 Report (New Reviewer)
Dear Authors,
I am very concerned regarding the methods you used to perform this study.
First of all data from 2010 should be renewed before publishing.
Second, six trees per species do not convince me at all. Further the design or your experiment is very questionable.
Presented results are very doubtful since the method part is very weak.
Next you do not present meaningful discussion part. It has to appear in your study.
Conclussions are rather weak. Tehy should hihglight the broader meaning of your study.
Regards
Author Response
Reviewer 4's five concerns are addressed as follows the CAPS after each concern:
1. First of all data from 2010 should be renewed before publishing. NO CHANGE as this concern is vague.
2. Second, six trees per species do not convince me at all. Further the design or your experiment is very questionable. NO CHANGE Certainly, the number of trees is limited, but the IRREC trees are all one cultivar, which greatly reduces variation among trees.
3. Presented results are very doubtful since the method part is very weak. ADDED "Nine topped and nine whole trees with bark..." and "Due to the generally smaller sizes of the IRREC trees (Table 2), IRREC wood sampling was based on stem disks instead of cores." Many statistics were eliminated due to previous reviewers' suggestions.
4. Next you do not present meaningful discussion part. It has to appear in your study. MINOR ADDITION "or via evapotranspiration by trees with intact foliage."
5. Conclussions are rather weak. Tehy should hihglight the broader meaning of your study. ADDED "..., which decreased with longer storage and increased seasonal temperature, relative humidity, and evapotranspiration."
This manuscript is a resubmission of an earlier submission. The following is a list of the peer review reports and author responses from that submission.
Round 1
Reviewer 1 Report
Evaluation of Field Drying for Enhancing Biomass Quality of Eucalyptus Logs and Trees for Biochar Production is a significative topic. This manuscript should be major-revised as follow before publication.
1. The title is suggested to be more briefly and clearly.
2. The abstract should show more briefly-clearly what problem you want to reviews and what contribution of the manuscript you want provide.
3. In the Introduction, the review for biochar utilization should further suppressed and the originality of this work should be more clearly stated after fully reviewing the related previous works.
4. The Fig 1 seems meaningless in the manuscript.
5. The results obtained in this work seem to be applicable only to evaluate field drying for enhancing biomass quality for biochar. The authors are recommended to state the wider applicability of their results.
6. Please check the language, grammar issues, and logicality throughout the manuscript.
Author Response
Our manuscript has had the following major revisions to address Reviewer 1's helpful suggestions:
- The title is suggested to be more briefly and clearly. - Title has been shortened.
- The abstract should show more briefly-clearly what problem you want to reviews and what contribution of the manuscript you want provide. - Has been revised
- In the Introduction, the review for biochar utilization should further suppressed and the originality of this work should be more clearly stated after fully reviewing the related previous works. - Intro has been broadened and expanded.
- The Fig 1 seems meaningless in the manuscript. - Figure 1 illustrates the drying process and has been retained
- The results obtained in this work seem to be applicable only to evaluate field drying for enhancing biomass quality for biochar. The authors are recommended to state the wider applicability of their results. - Applications have been broadened to other bioproducts; 12 references have been added.
- Please check the language, grammar issues, and logicality throughout the manuscript. - These checks have been performed.
Reviewer 2 Report
Review of the article: “Evaluation of Field Drying for Enhancing Biomass Quality of Eucalyptus Logs and Trees for Biochar Production in Florida, USA”
The topic of this research and output might be useful for the Forests readers. The article is very interesting and presents a rarely discussed but valuable topic regarding wood drying. The conducted work is worth to publish, however the article needs some corrections. The article requires refinement in terms of editing.
Title
In my opinion, the terms "in Florida, USA" should be deleted. Details of the location can be provided in the abstract or in the description of the research material in the section - Materials and Methods.
Abstract
Line 11: … Eucalyptus …
- Why is Eucalyptus capitalized?
Line 12: … 24 6- and …
- What does this record mean? There was a mistake in the record here.
Line 19: … 19 3+-year-old …
- What does this record mean? There was a mistake in the record here.
Line 22: It is advisable to give the dependencies in numbers or percentages. By how much was the MC reduction? This type of information increases the citation of the article.
Keywords: Eucalypts …
- or Eucalyptus?
Introduction
Line 68: Our specific main objectives were to determine the effects …
Use the impersonal form, e.g. The specific main objectives were to determine the effects …
Results and Discussion
- Please standardize the entries, because once it is e.g. 1 month, sometimes ~1 month.
Table 4: Why is the relative humidity of the air not given? The equilibrium moisture content of wood depends on the temperature and relative humidity of the air. And these are the parameters most often analyzed in relation to the moisture content of dried wood.
Line 250: Figure 5: EG (top) and EA (bottom) …
Should be rather Figure 5: EG (A) and EA (B) … Please correct.
Line 284: Figure 6: EG (A) and EA (B) MC …
- Please see comments for Line 250.
Conclusions
Please clearly specify the most important conclusions in points.

Author Response
Our revision addresses Reviewer 2's editorial and other suggestions as follows:
Title - has been shortened but retains "in Florida, USA" to allow reader immediate geographic recognition.
Abstract Line 11: Eucalyptus is capitalized and italicized to establish genus.
Line 12: "24" identifies the number of FORCE study sample trees.
Line 19: "19" states the number of IRREC study sample trees.
Line 22: Wording retained
Keywords: Changed to Eucalyptus
Introduction
Line 68: Changed to "The specific main objectives ..."
Results and Discussion
- The drying durations vary slightly.
Table 4: Because relative humidities at the study sites are high all year, they were not included.
Line 250: The (top) and (bottom) are retained because the figures are not labelled as A and B
Line 284: Same in Figure 6
Conclusions
Because this section is concise, points seem unnecessary.
Reviewer 3 Report
Dear Authors,
I have reviewed the paper titled: “Evaluation of Field Drying for Enhancing Biomass Quality of Eucalyptus Logs and Trees for Biochar Production in Florida, 3 USA". In my opinion, the aims of the paper are germane with “Forests” journal topic, however, in the present form, the paper doesn’t fit with the international scientific standards. The paper is written with a good English level. The contribution of this paper to the scientific knowledge is not acceptable and the main flaws present are highlighted in the text with various comments. I understand the difficult work done, but as a reviewer it is my duty to highlight the gaps in order to improve the research approach and its presentation to the international scientific community. Please I suggest to reconsider the paper following the suggestions and comments reported in the pdf attached.

Author Response
Our revised paper better matches international scientific standards.
As other reviewers have commented, our paper is a helpful contribution to the literature on field drying.
Its numerous revisions address many of the points nicely highlighted by Reviewer 3.
Reviewer 4 Report
The Authors have prepared an interesting paper related to the natural drying process of roundwood in the form of logs and logs along with tree crowns.
The paper presents results from 2010 - 2019 with spring and summer harvesting.
The methodology needs significant supplementation with an indication of the storage conditions of the harvested research material. The authors did not present data on weather changes during the study period and did not include them in the presentation of the research methodology. Their inclusion in the results is incorrect as they are part of the factor influencing the wood drying process.
. It is necessary to refer to the literature related to the water diffusion process in the studied species. No description of the change in hygroscopic equilibrium on drying speed. No discussion of the influence of the macroscopic and microscopic structure of wood on the speed of water propagation. The method of face preparation and the degree of bark reduction and the influence of wood surface on water diffusion are not indicated.
Preliminary results of moisture distribution are related to plantation growing conditions and take into account soil and moisture variations and fertilisation of the studied species.
The result of faster evaporation in summer seems natural and is due to a higher osmotic pressure difference. This is due to a change in the hygroscopic equilibrium of the wood with an increase in ambient temperature.
The authors should improve the results presented by properly formatting the tables and figures.
The reference to the pyrolysis process in the results should be supported by the requirements for the preparation of the wood raw material. There is a lack of development of a literature review for the effect of moisture content on the pyrolysis or combustion process.
Author Response
The methodology needs significant supplementation with an indication of the storage conditions of the harvested research material. - Figure 1 describes the field drying method for both studies, and all cores and disks went directly into a drying room as stated on Lines 147 and 205
The authors did not present data on weather changes during the study period and did not include them in the presentation of the research methodology. - Table 4 gives the differences between the FORCE study spring and summer seasons. Summers in Florida are very uniform, as illustrated by the data for three Florida locations in Table 4.
It is necessary to refer to the literature related to the water diffusion process in the studied species. No description of the change in hygroscopic equilibrium on drying speed. No discussion of the influence of the macroscopic and microscopic structure of wood on the speed of water propagation. - Literature has been added to cover these important points.
The method of face preparation and the degree of bark reduction and the influence of wood surface on water diffusion are not indicated - Figure 1 illustrates these circumstances.
Preliminary results of moisture distribution are related to plantation growing conditions and take into account soil and moisture variations and fertilisation of the studied species - Table 1, Lines 99-102, and Lines 185-188 describe the cultural conditions.
The result of faster evaporation in summer seems natural and is due to a higher osmotic pressure difference. This is due to a change in the hygroscopic equilibrium of the wood with an increase in ambient temperature. - These helpful points have been addressed in the discussion.
The authors should improve the results presented by properly formatting the tables and figures. - Reformatting has been done
The reference to the pyrolysis process in the results should be supported by the requirements for the preparation of the wood raw material. There is a lack of development of a literature review for the effect of moisture content on the pyrolysis or combustion process. - This literature has been expanded by 12 mostly recent references.
Round 2
Reviewer 1 Report
Accept
Reviewer 3 Report
Dear Authors,
Dear Authors,
I have reviewed the paper titled: “Evaluation of Field Drying for Enhancing Biomass Quality of Eucalyptus Logs and Trees for Biochar Production in Florida, USA". In my opinion, the aims of the paper are germane with “Forests” journal topic, however, also in the present form, the paper doesn’t fit with the international scientific standards. The paper is written with a good English level. The contribution of this paper to the scientific knowledge is not acceptable and the main flaws present are highlighted in the text with various comments, in particular referred to the number of trees for treatment, it is difficult to show statistics without a real variance. I understand the difficult work done, but as a reviewer it is my duty to highlight the gaps in order to improve the research approach and its presentation to the international scientific community. Please I suggest to reconsider the paper following the suggestions and please, detail in the reply your reasons.